# A Simple Yet Effective SVD-GCN for Directed Graphs

## Abstract

In this paper, we will present a simple yet effective way for directed Graph (digraph) Convolutional Neural Networks based on the classic Singular Value Decomposition (SVD), named SVD-GCN for digraphs. Through empirical experiments on node classification datasets, we have found that SVD-GCN has remarkable improvements in a number of graph node learning tasks and outperforms GCN and many other state-of-the-art graph neural networks.

## 1 Introduction

Recent decades have witnessed the success of deep learning in many domains due to its capabilities to effectively extract the latent information from data and efficiently capture the hidden patterns of data, no matter it is textual data or image data (Bronstein et al., 2017; Wu et al., 2020; Zhou et al., 2020b; Zhang et al., 2020; Atz et al., 2021). In recent years, graph representation learning has emerged and become one of the most prominent fields in deep learning and machine learning areas (Battaglia et al., 2018). Many real-world data are presented or structured in the form of graphs reflecting their intrinsic relations. For example, in a citation network, publications are represented as nodes while the edges represent the citation relationship. To better handle the complexity and capture the hidden information of graph-structured data, Graph Neural Networks (GNNs) have become an effective and useful tool for graph learning tasks. The underlying principle of GNNs is to transform graph data into a low-dimensional space while maintaining the structural information as much as possible (Wu et al., 2020).

Although GNNs have been successfully applied in many applications, existing literature primarily focuses on undirected graphs, despite a variety of tasks involving learning on directed graphs (digraphs) where the edges encode extra directional information. Examples include citation graphs (An et al., 2004), transportation prediction (Zhou et al., 2020a), website hyperlinks (Pei et al., 2020) In this paper, we particularly consider directed graphs, which include undirected graphs as special cases.

Generally, two classes of GNNs exist, i.e., spatial-based and spectral-based approaches. Spatial methods apply message passing (feature aggregation) from the neighbours of each graph node, while spectral methods apply convolution in the graph Fourier domain through eigen-decomposition (EVD) of the graph Laplacian. Spatial methods can be readily adapted to directed graphs by aggregating the features with directions. However, compared to spectral methods, spatial methods are less favored due to their inability to extract information at different frequencies. In fact, as shown in (Balcilar et al., 2020a;b; Nt & Maehara, 2019), many spatially-designed architectures are low-pass filters, thus failing to model high-frequency information that is often more useful.

On the other hand, it is not straightforward to generalize spectral methods to directed graphs because the asymmetric adjacency and Laplacian matrices do not provide orthonormal systems for signal decomposition (Wu et al., 2020; Tong et al., 2020a). One naive strategy is to ignore the directions by performing convolution on the symmetrized adjacency/Laplacian matrix (Kipf & Welling, 2017; Wu et al., 2019b). However, this result is in severe loss of information and often performs worse than specialized GNNs for directed graphs (as shown in the experiment section). Particularly, in (Ma et al., 2019), the directed graph convolution is performed on a symmetric Laplacian defined via the transition probability matrix. Furthermore, Tong et al. (2020a) extends the idea in (Ma et al., 2019) that only works for strongly connected graphs to general

graphs by adding a small teleport probability in the transition matrix. In addition, the paper enhances the performance by developing scalable receptive fields based on the idea of the inception network (Szegedy et al., 2016). The experiment results validate the efficacy of the specialized methods over the naive strategy of removing directions as well as many spatial methods.

Motivated by the success of the aforementioned methods for directed graphs, we wish to retain the superiority of the specialized spectral GNNs for directed graphs while avoiding carefully designed adjacency or Laplacian matrix. To this end, different from the existing approaches, we perform convolution over the spectral domain given by singular value decomposition (SVD) of the directed adjacency matrix (which we call SVD-GCN). It is worth highlighting that such a strategy can be applied to any structured matrix of a directed graph, not limited to the graph adjacency. In addition, we leverage graph (quasi)-framelets for multi-resolution analysis on directed graphs, which provides better modeling of both low-pass and high-pass information.

Our *contributions* of this paper are in four-fold:

1. To our best knowledge, this is the first attempt to introduce adjacency SVD for graph convolutions neural network. To better filter the graph signals in the spectral domain, we apply the quasi-framelet decomposition of the graph signals to enhance the performance and robustness of the SVD-GCN.

2. We theoretically prove that the dual orthogonal systems offered by SVD provide the guarantees of graph signal decomposition and reconstruction based on the spectral theory.

3. We investigate the way of scaling up the SVD-GCN for large graphs based on Chebyshev polynomial approximation by deriving fast filtering for singular values without explicitly conducting SVD.

4. The results of extensive experiments prove the effectiveness of node representation learning via the framelet SVD-GCNs and show the outperformance against many state-of-the-art methods for digraphs.

**Organizations.**   The paper is organized as follows. Section 2 is dedicated to introducing the relevant works on graph neural networks for directed graphs and reviewing the graph framelets/quasi-framelets developed in recent years, to pave the way for how this can be used in directed graphs. Section 3 introduces the theory of graph signal SVD and its combination with the graph framelets/quasi-framelets which leads to our proposed SVD-GCN schemes for directed graphs. In Section 4, comprehensive experiments are conducted to demonstrate the robustness and effectiveness of the proposed SVD-GCN and its performance against a wide range of graph neural networks model/algorithms in node classification tasks. Section 5 concludes the paper.

## 2   Related Works

In this section, we will present a summary of the several works regarding GCN for directed graphs and framelet-based convolutions.

### 2.1   GCN for Directed Graphs

Neural network was first applied to directed acyclic graphs in (Sperduti & Starita, 1997), and this motivated the early studies in GNNs. But the early research mostly focuses on Recurrent GNNs (RecGNN) (Wu et al., 2020). Later, the Graph Neural Network (GNN) concept was proposed and further elaborated in (Scarselli et al., 2008) and (Gori et al., 2005), that expands the application domain of existing neural networks even larger and GNNs can be implemented to process more graph-based data. Inspired by the success of Convolutional Neural Network (CNN) application in computer vision, researchers developed many approaches that can redefine the notion of graph's convolution, and all these approaches and methods are under the umbrella of Convolutional Graph Neural Network (ConvGNN) (Wu et al., 2020).

We have noted that ConvGNNs can be categorized into spatial-based and spectral-based approaches. The majority of them are spatial models, the spatial-based approaches utilize neighbour traversal methods to extract and concatenate features, and this is implemented via taking adjacency as transformation in general.

Researchers did improve models' abilities of features extraction by stacking many graph convolutional layers (Wu et al., 2020). However, this approach could cause the overfitting problem and feature dilution issue as the models are built deeper with more and more GCN layers (Tong et al., 2020b).

Recently, more attention has focused on the provision of learning from directed graphs. A novel approach, Directed Graph Convolutional Network (DGCN) propagation model was developed and presented to adapt to the digraph (Ma et al., 2019). The key idea is to redefine a symmetric so-called normalized Laplacian matrix for digraphs via normalizing and symmetrizing the transition probability matrix. DGCN does have better performance than the state-of-the-art spectral and spatial GCN approaches on directed graph datasets in semi-supervised nodes classification tasks, but it still has some limitations, such as high computational cost, high memory space requirement and its strongly connected digraph assumption (Ma et al., 2019).

In (Monti et al., 2018), the researchers proposed a GCN called MotifNet, which is able to deal with directed graphs by exploiting local graph motifs. It basically uses the motif-induced adjacencies, constructed convolution-like graph filters and applied attention mechanism, while the experimental results on the real data shows that MotifNet does have advantages in dealing with directed graphs and addressing the drawback of spectral GCNs application in processing graph data, without further increasing the computational cost.

In (Tong et al., 2020a), inspired by the Inception Network module presented in (Szegedy et al., 2016), researchers proposed the Digraph Inception Convolutional Networks (DiGCN) in which they design and develop scalable receptive fields and avoid those unbalanced receptive fields which are caused by the non-symmetric digraph (directed graph). Through experiments, DiGCN is proved that it's able to learn digraph representation effectively and outperforms mainstream digraph benchmarks' GCN convolution.

One of the common ideas used in all the approaches is to use heuristics to construct and revise the Laplacian matrix. For example, the recent work (Zhang et al., 2021) proposes a way to define the so-called magnetic Laplacian, as a complex Hermitian matrix, in which the direction information is encoded by complex numbers. The experimental results manifest that MagNet's performance exceeds all other approaches on the majority of the tasks such as digraph node classification and link prediction (Zhang et al., 2021).

### 2.2 Framelets and Quasi-Framelets

Framelet-based convolutions have been applied to graph neural networks and reveal superior performance in keeping node-feature-related information and graph geometric information, while framelet-based convolutions also have the advantage of fast algorithm in signal decomposition and reconstruction process (Yang et al., 2022).

Framelet-based convolution for signals defined on manifolds (Dong, 2017) has been recently applied for graph signals (Zheng et al., 2021). Instead of using a single modulation function $g(\cdot)$ as in (2), a group of modulation functions in spectral domain was used, i.e., the scaling functions in Framelet terms. This set of functions can jointly regulate the spectral frequency and is normally designed according to the Multiresolution Analysis (MRA) based on a set of (finite) filter bank $\eta = \{a; b^{(1)}, ..., b^{(K)}\} \subset l_0(\mathbb{Z})$ in spatial domain. Yang et al. (2022) further demonstrate that the MRA is unnecessary, and propose a sufficient identity condition of a group modulation functions, which is reflected in the following definition

**Definition 2.1** (Modulation functions for Quasi-Framelets). *We call a set of $K+1$ positive modulation functions defined on $[0, \pi]$, $\mathcal{F} = \{g_0(\xi), g_1(\xi), ..., g_K(\xi)\}$, a quasi-framelet if it satisfies the following identity condition*

$$g_0(\xi)^2 + g_1(\xi)^2 + \cdots + g_K(\xi)^2 \equiv 1, \quad \forall \xi \in [0, \pi] \tag{1}$$

*such that $g_0$ decreases from 1 to 0 and $g_K$ increases from 0 to 1 over the spectral domain $[0, \pi]$.*

Particularly $g_0$ aims to regulate the high frequency while $g_K$ aims to regulate the lower frequency, and the rest will regulate other frequencies between. The classic examples include the linear framelet and quadratic framelet functions (Dong, 2017; Zheng et al., 2021), and sigmoid and entropy quasi-framelet functions (Yang et al., 2022). For the convenience, we list two examples here:

*Linear Framelet Functions* (Dong, 2017):

$$g_0(\xi) = \cos^2(\xi/2); \quad g_1(\xi)\frac{1}{\sqrt{2}}\sin(\xi); \quad g_2(\xi) = \sin^2(\xi/2).$$

*Entropy Framelet Functions* (Yang et al., 2022):

$$g_0(\xi) = \begin{cases} \sqrt{1 - g_1^2(\xi)}, & \xi <= \pi/2 \\ 0, & \text{otherwise} \end{cases}; \quad g_1(\xi) = \sqrt{4\alpha\xi/\pi - 4\alpha\,(\xi/\pi)^2}; \quad g_2(\xi) = \begin{cases} \sqrt{1 - g_1^2(\xi)}, & \xi > \pi/2 \\ 0, & \text{otherwise} \end{cases}$$

where $0 < \alpha \leq 1$ is a hyperparameter that can be fine-tuned. Note, for $\alpha = 1$, $g_1^2(\pi\xi)$ is the so-called *binary entropy function*.

## 3 SVD-Framelet Decomposition

### 3.1 The Motivation to apply SVD-based Approach

Spectral graph neural networks have been proved powerful for graph tasks. This is based on the node Laplacian $\mathbf{L}$ or the 0-th order Hodge Laplacian shown in (Lim, 2020). Suppose $\mathbf{X}$ represents the node signals on the graph, then the base operation in spectral GNN is $\mathbf{Y} = \mathbf{L}\,\mathbf{X}$. For undirected graphs, performing SVD of a symmetric matrix is nearly equivalent to EVD of the matrix, while the only difference is the sign. Consider EVD of the normalized Laplacian matrix $\widehat{\mathbf{L}} = \mathbf{I} - \widehat{\mathbf{A}}$ where $\widehat{\mathbf{A}} = (\mathbf{D} + \mathbf{I})^{-1/2}(\mathbf{A} + \mathbf{I})(\mathbf{D} + \mathbf{I})^{-1/2}$ as $\widehat{\mathbf{L}} = \mathbf{U}\mathbf{\Lambda}\mathbf{U}^\top = \mathbf{U}(\mathbf{I} - \mathbf{\Sigma})\mathbf{U}^\top$, where $\widehat{\mathbf{A}} = \mathbf{U}\mathbf{\Sigma}\mathbf{U}^\top$. One can verify the eigenvalues of $\widehat{\mathbf{L}}$ are in $[0, 2]$ and the eigenvalues of $\widehat{\mathbf{A}}$ are in $[-1, 1]$. The eigenvalues of Laplacian can be interpreted as the frequencies of graph signals defined on the graph nodes.

The above standard graph spectral methods usually consider graph Laplacian, which is symmetric, positive semi-definite (for undirected graphs). With these aforementioned properties, the classic Fourier analysis can be extended to graph signal processing with a set of orthonormal bases naturally formed by the eigenvectors of the normalized graph Laplacian. However, it will be a different story in the case of directed graphs, as we no longer enjoy the benefit of the symmetric property of the Laplacian. Consequently, we provide an alternative strategy based on SVD of the adjacency matrix (which could be self-looped) which plays a key role in a typical spatial Graph neural network, relying on the basic operation: $\mathbf{Y} = \mathbf{A}\mathbf{X}$, where $\mathbf{A}$ is the adjacency matrix which is normally asymmetric for digraphs.

In fact, the adjacency matrix can be regarded as the graph shift operator that replaces the graph signal at one node with the linear combination of its neighbours (Gavili & Zhang, 2017; Sandryhaila & Moura, 2013). Regardless of whether the graph is undirected or directed, as long as the adjacency matrix is diagonalizable, we can always factor $\mathbf{A} = \mathbf{V}\mathbf{\Lambda}\mathbf{V}^{-1}$ and generalize the Fourier transform as $\widehat{\mathbf{x}} = \mathbf{V}^{-1}\mathbf{x}$. The adjacency matrix is diagonalizable for strongly connected directed graphs (van Dam & Omidi, 2018) and we may consider Jordan decomposition when such condition is violated (Sandryhaila & Moura, 2013). Notice that for general directed graphs, $\mathbf{V}^{-1} \neq \mathbf{V}^\top$ and also the $\mathbf{S}, \mathbf{V}$ are complex-valued

To avoid such issue, we consider the SVD of the shift operator, $\mathbf{A} = \mathbf{U}\mathbf{\Lambda}\mathbf{V}^\top$, which provides two sets of orthonormal bases $\mathbf{U}, \mathbf{V}$ with real-valued, positive singular values $\mathbf{\Lambda}$. By applying the shift operator to a graph signal $\mathbf{X}$, it can be interpreted as first decomposing the signal in terms of the bases defined by the columns of $\mathbf{V}$, which is followed by a scaling operation defined by $\mathbf{\Lambda}$. Then the signals are transformed via another set of bases defined by the columns of $\mathbf{U}$.

As the magnitude of $\mathbf{\Lambda}$ indeed means the "frequency", we can regulate $\mathbf{\Lambda}$ by e.g. a modulation function $g$ and define the following filtering

$$\mathbf{Y} = \sigma((\mathbf{V}g(\mathbf{\Lambda})\mathbf{U}^\top) \cdot g_\theta \circ (\mathbf{U}g(\mathbf{\Lambda})\mathbf{V}^\top\mathbf{X})). \tag{2}$$

Extra care should be paid that the smaller singular values mean noised signal components. A well-designed modulation function $g$ shall learn to regulate the "frequency" components in the graph signals $\mathbf{X}$ in conjunction with a filter $g_\theta$, (Chang et al., 2021).

### 3.2 The Definition of SVD-Framelets

The recent work on the undecimated framelets-enhanced graph neural networks (UFG) has enjoyed its great success in many graph learning tasks (Zheng et al., 2021). As UFG is built upon the spectral graph signal analysis framework by exploiting the power of multiresolution analysis provided by the classic framelet theory (Dong, 2017), it can only be applied to undirected graphs. On the other side, the classic framelet construction is very restrictive. To explore more meaningful signal frequency decomposition, the authors of (Yang et al., 2022) propose a more flexible way to construct framelets, which are called quasi-framelets.

How to explore the multiresolution lens for digraph signals is a valuable question to ask. This motivates us to look back on whether any classic signal analysis methods can be adopted for digraph signals.

Now consider a directed (homogeneous) graph $\mathcal{G} = (\mathcal{V}, \mathcal{E})$ with $N$ nodes and any graph signal $\mathbf{X}$ defined on its nodes. Suppose that $\mathbf{A} \in \mathbb{R}^{N \times N}$ denotes its adjacency matrix which is typically asymmetric and its in-degree and out-degree diagonal matrices $\mathbf{D}_1$ and $\mathbf{D}_2$. Now we will consider its self-looped normalized adjacency matrix $\widehat{\mathbf{A}} = (\mathbf{D}_1 + \mathbf{I})^{-1/2}(\mathbf{A} + \mathbf{I})(\mathbf{D}_2 + \mathbf{I})^{-1/2}$. Typically in the spatial graph neural networks, $\widehat{\mathbf{A}}$ is used to define the following convolutional layer

$$\mathbf{X}' = \widehat{\mathbf{A}}\mathbf{X}\mathbf{W}. \tag{3}$$

Now suppose we have the following SVD for the normalized (directed) adjacency matrix

$$\widehat{\mathbf{A}} = \mathbf{U}\mathbf{\Lambda}\mathbf{V}^\top, \tag{4}$$

where $\mathbf{U}$ contains the left singular vectors, $\mathbf{V}$ contains the right singular vectors, and $\mathbf{\Lambda} = \mathrm{diag}(\lambda_1, ..., \lambda_N)$ is the diagonal matrix of all the singular values in decreasing order. Taking (4) into (3) reveals that we are projecting graph node signals $\mathbf{X}$ onto the orthogonal system defined by the columns of $\mathbf{V}$, then reconstructing the signals on its *dual* orthogonal system defined by the columns of $\mathbf{U}$, with appropriate scaling given by the singular values $\mathbf{\Lambda}$. This is the place we can "filter" graph signals according to the dual orthogonal systems.

Inspired by the idea of applying undecimated framelets over the Laplacian orthogonal systems (i.e. spectral analysis), we will introduce applying framelets over the dual orthogonal systems defined by SVD.

For a given set of framelet or quasi-framelet functions $\mathcal{F} = \{g_0(\xi), g_1(\xi), ..., g_K(\xi)\}$ defined on $[0, \pi]$[1], see (Yang et al., 2022; Zheng et al., 2021). and a given multiresolution level $L\ (\geq 0)$, define the following framelet or quasi-framelet signal decomposition and reconstruction operators

$$\mathcal{W}_{0,L} = \mathbf{V}g_0(\frac{\mathbf{\Lambda}}{2^{m+L}})\cdots g_0(\frac{\mathbf{\Lambda}}{2^m})\Lambda^{\frac{1}{2}}\mathbf{V}^\top, \quad \mathcal{W}_{k,0} = \mathbf{V}g_k(\frac{\mathbf{\Lambda}}{2^m})\Lambda^{\frac{1}{2}}\mathbf{V}^\top, \text{for } k = 1, ..., K, \tag{5}$$

$$\mathcal{W}_{k,\ell} = \mathbf{V}g_k(\frac{\mathbf{\Lambda}}{2^{m+\ell}})g_0(\frac{\mathbf{\Lambda}}{2^{m+\ell-1}})\cdots g_0(\frac{\mathbf{\Lambda}}{2^m})\Lambda^{\frac{1}{2}}\mathbf{V}^\top, \quad \text{for } k = 1, ..., K, \ell = 1, ..., L. \tag{6}$$

and

$$\mathcal{V}_{0,L} = \mathbf{U}\Lambda^{\frac{1}{2}}g_0(\frac{\mathbf{\Lambda}}{2^m})\cdots g_0(\frac{\mathbf{\Lambda}}{2^{m+L}})\mathbf{V}^\top, \quad \mathcal{V}_{k,0} = \mathbf{U}\Lambda^{\frac{1}{2}}g_k(\frac{\mathbf{\Lambda}}{2^m})\mathbf{V}^\top, \text{for } k = 1, ..., K, \tag{7}$$

$$\mathcal{V}_{k,l} = \mathbf{U}\Lambda^{\frac{1}{2}}g_0(\frac{\mathbf{\Lambda}}{2^m})\cdots g_0(\frac{\mathbf{\Lambda}}{2^{m+\ell-1}})g_k(\frac{\mathbf{\Lambda}}{2^{m+\ell}})\mathbf{V}^\top, \quad \text{for } k = 1, ..., K, \ell = 1, ..., L. \tag{8}$$

We stack them as $\mathcal{W} = [\mathcal{W}_{0,L}; \mathcal{W}_{1,0}; ...; \mathcal{W}_{K,0}; \mathcal{W}_{1,1}; ...; \mathcal{W}_{K,L}]$ in column direction and $\mathcal{V} = [\mathcal{V}_{0,L}, \mathcal{V}_{1,0}, ..., \mathcal{V}_{K,0}, \mathcal{V}_{1,1}, ..., \mathcal{V}_{K,L}]$ in the row direction, then we have

**Theorem 3.1.** *The SVD-GCN layer defined by* (3) *can be implemented by a process of decomposition and reconstruction defined by two operators $\mathcal{W}$ and $\mathcal{V}$, i.e.,*

$$\mathbf{X}' = \widehat{\mathbf{A}}\mathbf{X}\mathbf{W} = \mathcal{V}(\mathcal{W}\mathbf{X}\mathbf{W}).$$

---

[1]The reason why we consider this domain as the most classic framelets are defined on $[0, \pi]$. This restriction can be removed for quasi-framelets.

*Proof.* Indeed we only need to prove that $\widehat{\mathbf{A}} = \mathcal{V}\mathcal{W}$. We will apply the identity property of the (quasi-)framelet functions, i.e., $\sum_{k=0}^{K} g_k^2(\xi) \equiv 1$. According to the definition of the matrices $\mathcal{W}$ and $\mathcal{V}$, we have

$$\mathcal{V}\mathcal{W} = \mathcal{V}_{0,L}\mathcal{W}_{0,L} + \sum_{\ell=0}^{L}\sum_{k=1}^{K}\mathcal{V}_{k,\ell}\mathcal{W}_{k,\ell} = \mathcal{V}_{0,L}\mathcal{W}_{0,L} + \sum_{k=1}^{K}\mathcal{V}_{k,L}\mathcal{W}_{k,L} + \sum_{\ell=0}^{L-1}\sum_{k=1}^{K}\mathcal{V}_{k,\ell}\mathcal{W}_{k,\ell}$$

$$=\mathbf{U}\mathbf{\Lambda}^{\frac{1}{2}}g_0(\frac{\mathbf{\Lambda}}{2^m})\cdots g_0(\frac{\mathbf{\Lambda}}{2^{m+L}})\mathbf{V}^{\top}\mathbf{V}g_0(\frac{\mathbf{\Lambda}}{2^{m+L}})\cdots g_0(\frac{\mathbf{\Lambda}}{2^m})\Lambda^{\frac{1}{2}}\mathbf{V}^{\top}$$

$$+ \sum_{k=1}^{K}\mathbf{U}\mathbf{\Lambda}^{\frac{1}{2}}g_0(\frac{\mathbf{\Lambda}}{2^m})\cdots g_0(\frac{\mathbf{\Lambda}}{2^{m+L-1}})g_k(\frac{\mathbf{\Lambda}}{2^{m+L}})\mathbf{V}^{\top}\cdot\mathbf{V}g_k(\frac{\mathbf{\Lambda}}{2^{m+L}})g_0(\frac{\mathbf{\Lambda}}{2^{m+L-1}})\cdots g_0(\frac{\mathbf{\Lambda}}{2^m})\Lambda^{\frac{1}{2}}\mathbf{V}^{\top} + \sum_{\ell=0}^{L-1}\sum_{k=1}^{K}\mathcal{V}_{k,\ell}\mathcal{W}_{k,\ell}$$

$$=\mathbf{U}\mathbf{\Lambda}^{\frac{1}{2}}g_0(\frac{\mathbf{\Lambda}}{2^m})\cdots g_0(\frac{\mathbf{\Lambda}}{2^{m+L-1}})\left(\sum_{k=0}^{K}g_k^2(\frac{\mathbf{\Lambda}}{2^{m+L}})\right)\cdot g_0(\frac{\mathbf{\Lambda}}{2^{m+L-1}})\cdots g_0(\frac{\mathbf{\Lambda}}{2^m})\Lambda^{\frac{1}{2}}\mathbf{V}^{\top} + \sum_{\ell=0}^{L-1}\sum_{k=1}^{K}\mathcal{V}_{k,\ell}\mathcal{W}_{k,\ell}$$

$$=\mathcal{V}_{0,L-1}\mathcal{W}_{0,L-1} + \sum_{l=0}^{L-1}\sum_{k=1}^{K}\mathcal{V}_{k,l}\mathcal{W}_{k,l} = \cdots = \mathcal{V}_{0,0}\mathcal{W}_{0,0} + \sum_{k=1}^{K}\mathcal{V}_{k,0}\mathcal{W}_{k,0} = \mathbf{U}\mathbf{\Lambda}^{\frac{1}{2}}\left(\sum_{k=0}^{K}g_k^2(\frac{\mathbf{\Lambda}}{2^m})\right)\Lambda^{\frac{1}{2}}\mathbf{V} = \mathbf{U}\mathbf{\Lambda}\mathbf{V} = \widehat{\mathbf{A}}.$$

This completes the proof. $\square$

### 3.3 SVD-Framelet Signal Decomposition and Reconstruction

To explore SVD-GCN layer from the graph signal point of view, we define the following graph SVD-framelets.

Suppose $\{(\lambda_i, \mathbf{u}_i, \mathbf{v}_i)\}_{i=1}^{N}$ are the singular values and singular vector triples for the normalized adjacency matrix $\widehat{\mathbf{A}}$ of graph $\mathcal{G}$ with $N$ nodes, such that $\{\lambda_i\}$ in decreasing order and $\mathbf{u}_i$ and $\mathbf{v}_i$ are columns of $\mathbf{U}$ and $\mathbf{V}$, respectively. Denote by $\beta_0^k(\xi) = g_k(\frac{\xi}{2^m})$ and $\beta_\ell^k(\xi) = g_k(\frac{\xi}{2^m})g_0(\frac{\xi}{2^{m-1}})\cdots g_0(\frac{\xi}{2^{m-\ell}})$ for $\ell = 1, 2, ..., L, k = 0, 1, ..., K$. For a graph $\mathcal{G}$, given a set of modulation functions $\mathcal{F} = \{g_0(\xi), g_1(\xi), ..., g_K(\xi)\}$, the forward SVD-framelets at scale levels $\ell$ are defined, for $k = 1, ..., K$, by

$$\phi_{0,p}(q) = \sum_{i=1}^{N}\sqrt{\lambda_i}\beta_L^0(\frac{\lambda_i}{2^L})\mathbf{v}_i(p)\mathbf{v}_i(q), \quad \psi_{0,p}^k(q) = \sum_{i=1}^{N}\sqrt{\lambda_i}\beta_0^k(\frac{\lambda_i}{2^0})\mathbf{v}_i(p)\mathbf{v}_i(q)$$

$$\psi_{\ell,p}^k(q) = \sum_{i=1}^{N}\sqrt{\lambda_i}\beta_\ell^k(\frac{\lambda_i}{2^\ell})\mathbf{v}_i(p)\mathbf{v}_i(q), \quad \ell = 1, 2, ..., L; \; 1, 2, ..., K. \tag{9}$$

and its corresponding backward SVD-framelets by

$$\overline{\phi}_{0,p}(q) = \sum_{i=1}^{N}\sqrt{\lambda_i}\beta_L^0(\frac{\lambda_i}{2^L})\mathbf{u}_i(p)\mathbf{v}_i(q), \quad \overline{\psi}_{0,p}^k(q) = \sum_{i=1}^{N}\sqrt{\lambda_i}\beta_0^k(\frac{\lambda_i}{2^0})\mathbf{u}_i(p)\mathbf{v}_i(q)$$

$$\overline{\psi}_{\ell,p}^k(q) = \sum_{i=1}^{N}\sqrt{\lambda_i}\beta_\ell^k(\frac{\lambda_i}{2^\ell})\mathbf{u}_i(p)\mathbf{v}_i(q), \quad \ell = 1, 2, ..., L; \; 1, 2, ..., K. \tag{10}$$

for all nodes $q, p$ and $\phi_{\ell,p}$ ($\overline{\phi}_{\ell,p}$) or $\psi_{\ell,p}^k$ ($\overline{\psi}_{\ell,p}$) are the low-pass or high-pass SVD-framelet translated at node $p$.

Similar to the standard undecimated framelet system (Dong, 2017), we can define two SVD-framelet systems as follows:

$$\text{SVD-UFS-F}_L(\mathcal{F};\mathcal{G}) := \{\phi_{0,p} : p \in \mathcal{V}\} \cup \{\psi_{\ell,p}^k : p \in \mathcal{V}, \ell = 0, ..., L\}_{k=1}^{K}, \tag{11}$$

$$\text{SVD-UFS-B}_L(\mathcal{F};\mathcal{G}) := \{\overline{\phi}_{0,p} : p \in \mathcal{V}\} \cup \{\overline{\psi}_{\ell,p}^k : p \in \mathcal{V}, \ell = 0, ..., L\}_{k=1}^{K}. \tag{12}$$

Then the signal transform $\mathbf{x}' = \widehat{\mathbf{A}}\mathbf{x}$ can be implemented via the SVD-Framelet systems as shown in the following theorem

**Theorem 3.2** (SVD-Framelet Transform). *Given the above definition of both forward and backward SVD-framelet systems, we have*

$$\mathbf{x}' = \sum_{p \in \mathcal{V}} \langle \phi_{L,p}, \mathbf{x} \rangle \overline{\phi}_{L,p} + \sum_{k=1}^{K} \sum_{\ell=0}^{L} \sum_{p \in \mathcal{V}} \langle \psi_{\ell,p}^k, \mathbf{x} \rangle \overline{\psi}_{\ell,p}. \tag{13}$$

*Proof.* The decomposition here is indeed the re-writing of $\mathbf{x}' = \widehat{A}\mathbf{x}$ in terms of columns of all the decomposition and reconstruction operators $\mathcal{W}$ and $\mathcal{V}$ under the modulation identity condition (1). □

This is to say that the transformed signal $\mathbf{x}'$ can be written as the linear combination of the backward SVD-framelet system with coefficients on the original signal on the forward SVD-framelet system. Thus the signal filtering can be conducted by filtering over the forward SVD-framelet coefficients $\langle \psi_{\ell,p}^k, \mathbf{x} \rangle$.

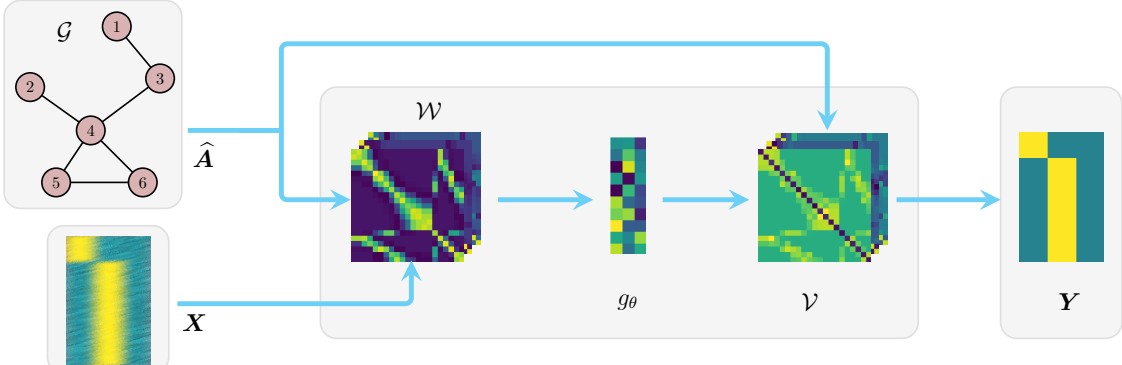

Figure 1: SVD Framelet Layer converts the input node feature $\mathbf{X}$ by using SVD framelet matrices $\mathcal{W}$ and $\mathcal{V}$ along with learnable filters $g_\theta$ to the new features $\mathbf{Y}$, as demonstrated in the simplified versions (14) and (15).

### 3.4 Simplified SVD-Framelet Filtering and the Model Architecture

Based on Theorem 3.1 and 3.2, for a demonstration, we define the following (simplified Level-1: corresponding to $L = 0$) SVD-Framelet filtering

$$\mathbf{Y} = \sigma \left( \sum_{k=0}^{K} (\mathbf{U}\Lambda^{\frac{1}{2}} g_k(\Lambda) \mathbf{V}^\top) \cdot g_\theta^k \circ (\mathbf{V} g_k(\Lambda) \Lambda^{\frac{1}{2}} \mathbf{V}^\top \mathbf{X} \mathbf{W}) \right) \tag{14}$$

where $g_\theta^k$'s are the filters corresponding to each modulation function $g_k$ individually, $\mathbf{W}$ is a learnable feature transformation weight and $\sigma$ is an activation function.

It is not necessary to transform the signal into the backward SVD-framelet space. We may consider the following signal transformation within the forward SVD-framelet space as the following way

$$\mathbf{Y} = \sigma \left( \sum_{k=0}^{K} (\mathbf{V}\Lambda^{\frac{1}{2}} g_k(\Lambda) \mathbf{V}^\top) \cdot g_\theta^k \circ (\mathbf{V} g_k(\Lambda) \Lambda^{\frac{1}{2}} \mathbf{V}^\top \mathbf{X} \mathbf{W}) \right) \tag{15}$$

in which $\mathbf{U}$ is replaced by $\mathbf{V}$.

It is not hard to write their counterparts for level 2. We call scheme (14) SVD-Framelet-I (including multiple levels) and (15) (including multiple levels) SVD-Framelet-II, however in the experiment part, we mainly focus on SVD-Framelet-I.

Figure 1 shows the information flow for one SVD framelet layer. First, the SVD will be conducted on the graph (normalized) adjacency matrix $\widehat{\mathbf{A}}$ to produce all the framelet matrices $\mathcal{W}$ (primary) and $\mathcal{V}$ (dual) up

to a given level $L$, then the primary framelet matrices $\mathcal{W}$ will be applied to the input node signal matrix $\mathbf{X}$, followed by the learnable filters on each node across all the channels, then the dual framelet matrices $\mathcal{V}$ will bring the signal back to the transformed signal domain, i.e., $\mathbf{X}'$, which will be pipelined to the next layers or downstream tasks. When multiple SVD layers are used in the final architecture, all the primary and dual framelet matrices are shared through all the SVD layers.

### 3.5 Faster filtering for large graphs based on Chebyshev polynomials

For large graphs, performing SVD on the adjacency matrix can be costly. We thus consider an approximated filtering based on Chebyshev polynomials. We will take another strategy to realize the idea presented in (Onuki et al., 2017) and to derive the fast filtering for singular values. Suppose that for the normalized adjacency matrix $\widehat{\mathbf{A}} = \mathbf{U}\mathbf{\Lambda}\mathbf{V}^\top = \widehat{\mathbf{A}}\mathbf{V}\mathbf{V}^\top$ and $\widehat{\mathbf{A}}^\top\widehat{\mathbf{A}} = \mathbf{V}\mathbf{\Lambda}^2\mathbf{V}^\top$. In other words, the columns of $\mathbf{V}$ give the eigenvector systems of $\widehat{\mathbf{A}}^\top\widehat{\mathbf{A}}$ for which we can conduct framelet analysis as done for Laplacian matrix.

For a given set of framelet or quasi-framelet functions $\mathcal{F} = \{g_0(\xi), g_1(\xi), ..., g_K(\xi)\}$ defined on $[0, \pi]$, as for (5) - (6), we define the following framelet or quasi-framelet signal decomposition operators (without confusion we use the same notation)

$$\mathcal{W}_{0,L} = \mathbf{V}g_0(\frac{\mathbf{\Lambda}^2}{2^{L+m}}) \cdots g_0(\frac{\mathbf{\Lambda}^2}{2^m})\mathbf{V}^\top, \quad \mathcal{W}_{k,0} = \mathbf{V}g_k(\frac{\mathbf{\Lambda}^2}{2^m})\mathbf{V}^\top, \text{for } k = 1, ..., K, \tag{16}$$

$$\mathcal{W}_{k,\ell} = \mathbf{V}g_k(\frac{\mathbf{\Lambda}^2}{2^{m+\ell}})g_0(\frac{\mathbf{\Lambda}^2}{2^{m+\ell-1}}) \cdots g_0(\frac{\mathbf{\Lambda}^2}{2^m})\mathbf{V}^\top, \quad \text{for } k = 1, ..., K, \ell = 1, ..., L. \tag{17}$$

Note that we have $\mathbf{\Lambda}^2$ inside all $g$'s but not the extra term $\mathbf{\Lambda}^{\frac{1}{2}}$.

To avoid any explicit SVD decomposition for $\mathbf{V}$, we consider a polynomial approximation to each modulation function $g_j(\xi)$ ($j = 0, 1, ..., K$). We approximate $g_j(\xi)$ by Chebyshev polynomials $\mathcal{T}_j^n(\xi)$ of a fixed degree $n$ where the integer $n$ is chosen such that the Chebyshev polynomial approximation is of high precision. For simple notation, in the sequel, we use $\mathcal{T}_j(\xi)$ instead of $\mathcal{T}_j^n(\xi)$. Then the new SVD-framelet transformation matrices defined in (16) - (17) can be approximated by, for $k = 1, ..., K, \ell = 1, ..., L$,

$$\mathcal{W}_{0,L} \approx \mathcal{T}_0(\frac{1}{2^{L+m}}\widehat{\mathbf{A}}^\top\widehat{\mathbf{A}}) \cdots \mathcal{T}_0(\frac{1}{2^m}\widehat{\mathbf{A}}^\top\widehat{\mathbf{A}}), \quad \mathcal{W}_{k,0} \approx \mathcal{T}_k(\frac{1}{2^m}\widehat{\mathbf{A}}^\top\widehat{\mathbf{A}}), \tag{18}$$

$$\mathcal{W}_{k,\ell} \approx \mathcal{T}_k(\frac{1}{2^{m+\ell}}\widehat{\mathbf{A}}^\top\widehat{\mathbf{A}})\mathcal{T}_0(\frac{1}{2^{m+\ell-1}}\widehat{\mathbf{A}}^\top\widehat{\mathbf{A}}) \cdots \mathcal{T}_0(\frac{1}{2^m}\widehat{\mathbf{A}}^\top\widehat{\mathbf{A}}), \tag{19}$$

Then there is no need for SVD of the adjacency matrix to calculate all the framelet matrices. So a new simplified one scale level SVD-Framelet-III (corresponding to $L = 0$) can be defined as

$$\mathbf{Y} = \sigma\left(\widehat{\mathbf{A}}\sum_{k=0}^{K}\mathcal{W}_{k,0}^\top \cdot g_\theta^k \circ (\mathcal{W}_{k,0}\mathbf{X}\mathbf{W})\right) \tag{20}$$

Similarly, the version of multiple levels can be easily written out.

## 4 Experiments

Our main purpose is to demonstrate the proposed SVD-GCN is powerful in assisting graph learning. We will evaluate our model in various graph learning tasks, including node classification, graph feature denoising, and applications to larger-scale graph data. The experiment code can be found at `https://github.com/ThisIsForReview/SVD-GCN` for review.

### 4.1 Datasets and Baselines

**Datasets** We utilize several digraph datasets from Python package `Torch_Geometric` `https://pytorch-geometric.readthedocs.io/` datasets in the experiments including: `cora_ml`, `citeseer`, and

Table 1: Statistics of the datasets

| Datasets | #Nodes | #Edges | #Classes | #Features |
|---|---|---|---|---|
| cora_ml (Bojchevski & Günnemann, 2017) | 2,995 | 8,416 | 7 | 2,879 |
| citeseer (Yang et al., 2016) | 3,312 | 4,715 | 6 | 3703 |
| citeseer_full (Chen et al., 2018) | 3,327 | 3,703 | 6 | 602 |
| amazon_photo (Shchur et al., 2018) | 7,650 | 143,663 | 8 | 745 |
| amazon_cs (Shchur et al., 2018) | 13,752 | 287,209 | 10 | 767 |
| cora_full (Bojchevski & Günnemann, 2018) | 19,793 | 65,311 | 70 | 8,710 |

citeseer_full which are citation networks, as well as the Amazon Computers and Amazon Photo co-purchase networks: amazon_photo and amazon_cs, see https://github.com/EdisonLeeeee/GraphData.

Here are the brief descriptions of the datasets we use to conduct the experiments and Table 1 summarizes the statistics of five datasets:

- cora_ml (Bojchevski & Günnemann, 2017): Cora is a classic citation network dataset while Cora_ml is a small subset of the dataset that (Bojchevski & Günnemann, 2017) extracted from the entire original network Cora dataset and cora_ml is also a directed network dataset, which means that the edge between all pairs of nodes is directed, ie. A pointed to B means that A cited B.

- citeseer (Yang et al., 2016) & citeseer_full (Chen et al., 2018): Citeseer is also a citation network dataset whose nodes represent documents and paper while edges are citation links. Citeseer_full is extracted from the same original network dataset as citeseer that nodes represent documents and edges represent citation links. The only difference is that Citeseer_full dataset's split type is full, which means that except the nodes in the validation and test sets, all the rest of nodes are used in training set.

- amazon_photo & amazon_cs (Shchur et al., 2018): The Amazon Computers and Amazon Photo network datasets are both extracted from Amazon co-purchase Networks. The nodes represent the goods and edges represent that the two nodes(goods) connected are frequently bought together, and the product reviews are bag-of-words node features.

Table 2: Results for Classification Accuracy (%): Part Results from (Tong et al., 2020a)

| Models | cora_ml | citeseer | citeseer_full | amazon_photo | amazon_cs |
|---|---|---|---|---|---|
| ChebNet | 64.02±1.5 | 56.46±1.4 | 62.29±0.3 | 80.91±1.0 | 73.25±0.8 |
| GCN | 53.11±0.8 | 54.36±0.5 | 64.71±0.5 | 53.20±0.4 | 60.50±1.6 |
| SGC | 51.14±0.6 | 44.07±3.5 | 56.56±0.4 | 71.25±1.3 | 76.17±0.1 |
| APPNP | 70.07±1.1 | 65.39±0.9 | 67.53±0.4 | 79.37±0.9 | 63.16±1.4 |
| InfoMax | 58.00±2.4 | 60.51±1.7 | 72.93±1.1 | 74.40±1.2 | 47.32±0.7 |
| GraphSAGE | 72.06±0.9 | 63.19±0.7 | 65.18±0.8 | 87.57±0.9 | 79.29±1.3 |
| GAT | 71.91±0.9 | 63.03±0.6 | 66.67±0.4 | 89.10±0.7 | 79.45±1.5 |
| UFG/QUFG | 76.98±5.3 | 64.70±1.7 | 80.44±1.3 | 80.98±1.5 | 77.32±4.3 |
| DGCN | 75.02±0.5 | 66.00±0.4 | 78.35±0.3 | 83.66±0.8 | OOM |
| SIGN | 64.47±0.9 | 60.69±0.4 | 77.44±0.1 | 74.13±1.0 | 69.40±4.8 |
| DiGCN-PR | 77.11±0.5 | 64.77±0.6 | 74.18±0.7 | OOM | OOM |
| DiGCN-APPR | 77.01±0.4 | 64.92±0.3 | 74.52±0.4 | 88.72±0.3 | 85.55±0.4 |
| **SVD-GCN (Ours)** | **78.84±0.29** | **66.15±0.39\*** | **80.95±0.36** | **88.76±0.21** | **85.55±0.31** |
| DiGCN-APPR-IB | 80.25±0.5 | 66.11±0.7 | 80.10±0.3 | **90.02±0.5** | **85.94±0.5** |
| **SVD-GCN-IB (Ours)** | **81.11±0.24** | 64.26±0.77 | **83.12±0.68** | 89.38±0.48 | 85.03±0.37 |

**Baseline Models** We will compare our model to twelve state-of-the-art models including spectral-based GNNs such as ChebNet (Defferrard et al., 2016), GCN (Kipf & Welling, 2017), APPNP (Klicpera et al., 2019), SGC (Wu et al., 2019a) and InfoMax (Veličković et al., 2019); spatial-based GNNs having GraphSAGE

(Hamilton et al., 2017) and GAT (Veličković et al., 2018); Graph Inception including SIGN (Rossi et al., 2020); Digraph GNNs containing DGCN; Digraph Inception having DiGCN-PR (Tong et al., 2020a), DiGCN-APPR (Tong et al., 2020a) and DiGCN-APPR-IB (Tong et al., 2020a).

We also conducted experiments for UFG/QUFG by using the Linear (Zheng et al., 2021) and Entropy (Yang et al., 2022) framelet functions. As UFG/QUFG is designed for undirected graphs, we will simply convert a directed graph to an undirected graph by adding the reversed edges.

At the time of preparing the draft, we are not aware the recently published paper (Zhang et al., 2021) on MagNet convolutional network for directed graphs. Thus we do not include it here for comparison. Nevertheless, we highlight that the number of parameters used in MagNet is double the size compared to our model and thus results in (Zhang et al., 2021) are not comparable. Further, the standard deviation of MagNet is much larger compared to our model, indicating less model robustness.

**Setup** We design our SVD-Framelet-I model (14) with one convolution layer plus a fully connected linear layer for learning the graph embedding, the output of which is proceeded by a softmax activation for final prediction. Most hyperparameters are set to default in our program, except for learning rate, weight decay, and hidden units in the layers. We conduct a grid search for fine-tuning on these hyperparameters from the pre-defined search space. All the models including those compared models that we need re-conduct are trained with the ADAM optimizer. The maximum number of epochs is basically 200.

**Hardware** Most of the experiments run in PyTorch 1.6 on NVIDIA® Tesla V100 GPU with 5,120 CUDA cores and 16GB HBM2 mounted on an HPC cluster and some experiments run with PyTorch 1.8 on PC with Intel(R) Core(TM) i5-8350U CPU@1.90GHz with 64 bits Window 10 operating system and 16GB RAM.

## 4.2 Graph Node Classification

We conduct our experiments on dataset `cora_ml`, `citeseer`, `cites eer_full`, `amazon_cs` and `amazon_photo`. In this set of experiments, the hyperparameters are searched in the following ways: 20 nodes per classes are randomly selected for training with 500 nodes as validation and the rest in testing; the basic epoch = 200 (but for `citeseer`, it is 500); we use the two-level framelets (corresponding to $L = 1$), the scale in framelets (i.e. $s$ in $s^l$ to replace $2^l$) is tested for 1.1, 1.5 and 2.0; the number of hidden features = 16, or 32, or 64; the dropout ratio = 0.1, 0.3, or 0.6; the framelet modulation function is either `Linear` or `Entropy` with hyperparameter $\alpha = 0.1, 0.3, 0.5, 0.7$ and 0.9. We find that both `Linear` and `Entropy` are comparable, thus the reported results are based on the `Linear` framelets. The network architecture consists of one SVD framelet layer (SVD-Framelet-I or SVD-Framelet-II for `citeseer`) and a fully connected linear layer to the softmax output layer.

The whole experimental results are summarized in Table 2. Except for `citeseer_full`, we copied the results from (Tong et al., 2020a) for convenience. The darker blue colour of the result cell represents the higher accuracy rate the approach generates using that dataset (within that column). Compared with all the other state-of-the-art baseline methods, the proposed method obtains the best performance or comparable performance, see all the rows above the two bottom rows. For datasets `cora_ml`, `citeseer` and `citeseer_full`, more than 1% gains have been obtained by such a simple convolution layer. For two Amazon datasets, the SVD-GCN is comparable to the most state-of-the-art model DiGCN-APPR. For `citeseer` data, we use the simplified model SVD-Frameelet-II (15). We also note that for the `citeseer_full` dataset, the SVD-GCN has a gain of more than 6% accuracy over the state-of-the-art DiGCN-PR and DiGCN-APPR (Tong et al., 2020a).

Tong et al. (2020a) further present the Digraph Inception Convolutional Networks (DiGCN-APPR-IB) which utilizes digraph convolution and $k$th-order proximity to achieve larger receptive fields and learn multi-scale features in digraphs. We also adapt their strategy and combine their architecture in our SVD-Framelet-I framelet, resulting in SVD-GCN-IB. The combined model achieves similar performance as the original DiGCN-APPR-IB, see the last two rows in Table 2. The results demonstrate that increasing receptive fields does improve the performance of both DiGCN and SVD-GCN.

For each dataset, we also report the best of either UFG (Linear) or QUFG (Entropy) for convenience. Here the purpose is to quickly compare SVD-GCN against UFG/QUFG. We have observed that UFG/QUFG

Table 3: Results between SVD-Framelet-III and GCN over `cora_full`

| NoHiddenUnits | GCN | **SVD-GCN (Ours)** |
|:---:|:---:|:---:|
| 64 | 52.54±0.41 | **53.52±0.33** |
| 128 | 53.75±0.17 | **56.17±0.16** |
| 256 | 54.33±0.33 | **57.30±0.21** |

generally performs well, even better than some GNNs designed for directed graphs. This further demonstrates the benefit of using multiple-scale decomposition in GNNs.

### 4.3 Larger Dataset Experiment

In this experiment, we would like to demonstrate the fast SVD-Framelet-III introduced in Section 3.5. All the experiments will be conducted on the dataset `cora_full`, see `https://github.com/EdisonLeeeee/GraphData`. `cora_full`, introduced in (Bojchevski & Günnemann, 2018), is the full extension of the smaller citation network dataset `cora_ml` and it consists of 19,793 nodes and 65,311 edges with node feature dimension of 8,710. The number of node classes is 70. This is a quite large graph for most graph neural networks. In fact, for graphs beyond 15K nodes, we had to revert to slow training on the CPU since the data did not fit on the GPU memory (12GB).

To get a sense of how reliable and robust our simplified fast SVD-Framelet-III based on the Chebyshev polynomial approximation is, we conduct the experiment only against the manageable benchmark GCN. In fact, for many state-of-the-art algorithms like DiGCN-PR or DiGCN-APPR, we failed to make the program run on the CPU.

Our experimental results are reported in Table 3. We follow the similar setting as the first set of experiments in Section 4.2: we choose 20 nodes per class (in total 70 classes), 500 random nodes for validation and the rest are used as testing nodes. This time, we fixed the framelet scale to $s = 1.1$ with the `Linear` framelet modulation functions and the dropout ratio to 0.1 with activation function `relu` applied to the output from the SVD framelet layer. The overall network architecture consists of one SVD framelet layer and a fully connected linear layer to the output softmax layer. As the dimension of the node feature is almost 8,710, we tested three different choices of the hidden unit size from 64, 128 and 256. Each experiment was run 10 replicates and each replicate runs 200 epochs with a fixed learning rate 0.005. We report the average test accuracy and their std. The experiment shows that the number of hidden unit - 128 is appropriate for this dataset and that the SVD-GCN has performance gains of 1 - 3% more in all the cases.

### 4.4 Denoising Capability and Robustness

We also conducted an experiment to assess the robustness of the proposed SVD-Framelet GCNs against the state-of-the-art method DiGCN-APPR. The experiments are conducted on the benchmark dataset `cora_ml` for convenience. The data features used in (Tong et al., 2020a) are normalized data with values ranging between 0.0 and 1.0. For the purpose of testing denoising capability of the two models SVD-Framelet-I and DiGCN-APPR, we randomly inject Gaussian noises of mean 0 and std (noise levels) ranging from 0.01 to 5. We report the results for noise levels 0.01, 0.05, 0.1, 0.5, 1.0, and 5.0 in Table 4. We did not report the results for DiGCN-APPR for larger noise levels due to its poor figures. The darker color in Table 4 represents the higher accuracy rate which further demonstrates the better denoising capability as it still can maintain a relatively high accuracy under different degrees of noise attack. If the accuracy rate is smaller than 40%, then the result cell will not be colored since that result is quite low and not comparable. From Table 4, it is obvious that when the noise level is larger than 0.01, the DiGCN-APPR method's accuracy rate has dropped dramatically, meanwhile SVD-GCN method still maintains a relatively high accuracy under the noise level of 5.0, and the result is even higher than DiGCN-APPR method's accuracy at the noise level of 0.01.

**Sensitivity analysis:** It is evident that DiGCN-APPR almost fails in denoising data, or in other words, it is quite easy to be attacked by noises. For example, the test accuracy degrades almost 24% from the noise-free case to the case with 0.01 noise level. However, SVD-GCN has better denoising capability, which benefits

Table 4: Denoising Capability Comparions between SVD-Framelet-I and DiGCN-APPR over `cora_ml`

| NoiseLevel | DiGCN-APPR (Tong et al., 2020a) | **SVD-GCN (ours)** |
|---|---|---|
| $\sigma = 0.0$ | 77.01±0.40 | 78.84±0.29 |
| $\sigma = 0.01$ | 53.39±0.61 | 76.04±0.51 |
| $\sigma = 0.05$ | 35.72±0.80 | 71.04±0.50 |
| $\sigma = 0.1$ | 34.08±1.34 | 68.25±0.57 |
| $\sigma = 0.5$ | 30.40±1.92 | 67.37±0.47 |
| $\sigma = 1.0$ | — | 66.38±0.80 |
| $\sigma = 5.0$ | — | 59.63±0.74 |

from the framelet decomposition over the SVD "frequency" domain and being filtered in its learning process. The experimental results demonstrate SVD-GCN's performance consistency and robustness to the larger noise levels. From Figure 2, we can observe that, at the noise level of 50, the test accuracy is still above 50% with acceptable standard deviation, which further proves its robustness and stability when encountering much larger noise attack.

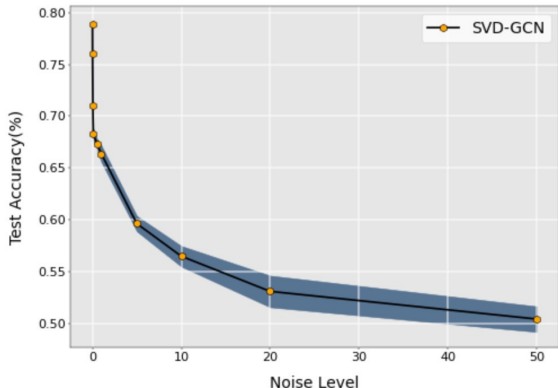

Figure 2: Node attribute perturbation analysis on `cora_ml` dataset

## 5    Conclusions

In this paper, we explored the application of framelets over the dual orthogonal systems defined by singular vectors from singular value decomposition (SVD) for graph data and thus proposed a simple yet effective SVD-GCN for directed graphs. The successful improvement from SVD-GCN benefits from the application of the graph SVD-framelets in transforming and filtering directed graph signals. The experimental results manifest that the SVD-GCN outperforms all the baselines and the state-of-the-art on the five commonly used benchmark datasets, which further proves that the proposed SVD-GCN's performance in directed graph learning tasks is remarkable.The sensitivity analysis also demonstrates that this novel approach has strong denoising capability and it is robust to high-level noise attack on node features. It has also been proved by experiments that the new fast SVD-GCN is convincingly accurate, reliable and appropriate in dealing with large-scale graph datasets.

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
