# OpenReview forum: "A Simple Yet Effective SVD-GCN for Directed Graphs"
_TMLR — Rejected by TMLR_

### Review · Reviewer_9MKT · 2022-05-24

**Summary Of Contributions:**

The main contribution of this paper is a spectral-based graph neural network for directed graphs. The authors base their solution upon the theory of quasi-framelet deccomposition. The authors also propose fast approximations to the proposed SVD-GCN. Extensive experiments on node classification show the effectiveness of the proposed SVD-GCN.

**Requested Changes:**

Weaknesses 3-5 in the experimental evaluations. If these weaknesses are addressed, this paper would be stronger.

**Strengths And Weaknesses:**

# Pros
1. The proposed method is backed up with theory in graph signal processing. I have no specific expertise in quasi-framelets, and thus I cannot evaluate the soundness of Sec. 3.2 and 3.3. However, Sec. 3.1 and 3.4 are largely sound.
2.

# Cons
1. I have no special expertise in spectral-based graph signal processing, and more specifically, I have no expertise on the concepts of quasi-framelets. Thus, I would appreciate a little more on the backgrounds.
2. Please replace all \cite to \citep. The existing citation style makes the paper hard to read.
3. It is a little strange to consider directed graphs only. In practice, what people usually do to directed graphs is to ignore the direction of edges and treat it as an undirected graph. This practice usually leads to better accuracy. For example, if we consider undirected versions of Cora-ml and Citeseer, it is common for simple models e.g. GCN and GraphSAGE to reach >80% for cora and >70% for citeseer. Thus, the proposed model may have limited practical use. The authors may want to show performances of baseline models with all edge directions removed.
4. Some experimental details are strange. For example, it is surprising that the authors call Cora-full with 20K nodes as "large-scale". In fact, graphs with much more nodes (e.g. Reddit , OGB-products, OGB-papers100M with millions of nodes) are used to evaluate GNNs (See OGB), which makes cora-full not "large-scale" enough. Also, it is strange that Cora-full cannot fit in a Tesla V100 with 16GB memory. From my experience, I was able to fit OGB-products with 2.45M nodes in a V100. Thus, I expect a more convincing justification about the "large-scale experiments".
5. I would like to see analyses on the parameters $K, L$ in Equations 7, 8 that are related to the ability of SVD-GCN.

---

> ### Author Response · Authors · 2022-06-20
> **Sorry :-) We are reseding our response to your review.  Our apologies for wrongly copying the comments.**
>
> 1. First, thank you very much for pointing out the citation style issue. It is our negligence when we convert from other styles to TMLR style. Now this has been fixed.
>
> 2. Regarding the strategy in dealing with directed graphs in GNNs, we dont agree with your comments "In practice, what people usually do to directed graphs is to ignore the direction of edges and treat it as an undirected graph. This practice usually leads to better accuracy.'' In your examples, cora is indeed an undirected graph. That is why most papers on directed graph NNs including ours here do not use this dataset. For citeseer, according to the literature (see our Table 2), the classic GCN gives only 54.36% accuracy while GraphSAGE gives 63.19%, but all GNNs for directed graphs like DGCN etc have 3% more improvement, including SVD-GCN with 66.15\%. This means the strategy taking care of directed information normally does better. Hence your conclusion "Thus, the proposed model may have limited practical use'' cannot stand.
>
> 3.  We agree with you that it is not appropriate to call a graph with 20K nodes as large-scale.  Given the fact that the first version SVD-GCN model relies on SVD which is computationally intense for large-scale graphs, we have proposed a fast version in Section 3.5 without explicitly conducting SVD. This will make the complexity roughly similar to the classic GCN which is sustainable for large-scale graphs. Given the computational resources we have (we are not in big company), we conducted only one experiment on cora-full.  In the rest of these four weeks review circle, we will try our best to conduct example from OGB like Reddit.
>
> 4.  Given the review time constraint, we are trying our best to analyze the  parameters like $L$, layers etc.  However, our experience and our reading from is that a value of $L=2$ is good enough in most case.

---

> > ### Author Response · Authors · 2022-06-20
> > **Correction to the last sentence**
> >
> >  However, our experience and our reading from UFG/QUFG papers is that a value of $L=2$ is good enough in most cases.

---

> > ### Comment · Reviewer_9MKT · 2022-06-20
> > **Response to author rebuttals.**
> >
> > I would like to briefly respond to the two points stated in the author response.
> >
> > 1. **Motivation on the directed graphs**. First, I would like to apologize on my mistake made about the "cora_ml" dataset. I confused it with the famous cora dataset (used in GCN, with 2708 nodes and 5429 edges). Results on cora_ml is fine.
> >
> > However, results on citeseer is strange. Specifically, the original GCN paper (https://arxiv.org/pdf/1609.02907.pdf) reported to have 70.6% on citeseer, using **undirected** graphs. However, in your paper, you report GCN to have ~56%, while the proposed SVD-GCN has ~66%. Please clarify the large gap between your GCN and the GCN in the original paper. Also, if using undirected citeseer leads to 70%, while SVD-GCN achieves 66%, the motivation of using directed graphs may be at doubt.
> >
> > 2. **Large-scale experiments**. First, I would like to point out that it does not require a big company to be able to run GNN on larger datasets such as Reddit/OGB-products. For example, for the reddit dataset (240k nodes), even if we adopt full-graph training (i.e. loading the whole dataset on GPU), it would take ~12GB GPU memory, which I don't think is very expensive (even a desktop RTX 3060 has 12GB memory). This cost can be further cut down by adopting mini-batch & sampling techniques. Therefore, cora_full is not even close to large-scale, and that running graphs with at least 200k nodes is not a very hard requirement.
> >
> > Second, scalability is an important aspect in real-world scenarios. If your method is not scalable (e.g. cannot perform single GPU training with only 20k nodes) compared to commonly used variants like GraphSAGE, then the proposed method is still limited.
> >
> > 3. Results on parameter study is relatively minor by comparison to the previous points. It is OK not to deal with it at present.

---

> > > ### Author Response · Authors · 2022-06-20
> > > **Clarification**
> > >
> > > The results for citeseer with GCN came from https://proceedings.neurips.cc/paper/2020/file/cffb6e2288a630c2a787a64ccc67097c-Paper.pdf. See Tables 2 and 3 in the paper.

---

> > > ### Author Response · Authors · 2022-06-20
> > > **Clarification 2**
> > >
> > > We dont believe two sets of citeseers data are same.   In https://arxiv.org/pdf/1609.02907.pdf, the dataset contain 3327 nodes, 4732 edges and features 3703.  While both https://proceedings.neurips.cc/paper/2020/file/cffb6e2288a630c2a787a64ccc67097c-Paper.pdf and our paper use the citeseer dataset from  https://github.com/EdisonLeeeee/GraphData with nodes 3,312, edges 4,715 and feature 3703.

---

> > > > ### Comment · Reviewer_9MKT · 2022-06-20
> > > > **Apologies for the rash conclusion, and updates after checking.**
> > > >
> > > > My apologies for my jumping to conclusions. Please give me some time to check whether they are indeed the same or not.
> > > >
> > > > ==========UPDATE===============
> > > >
> > > > I personally checked three citeseers.
> > > > 1. The default version in PyG and DGL, which is undirected, has 3327 nodes, 4732 edges, 3703 features. The statistics correspond with the citeseer used in Planetoid (https://github.com/kimiyoung/planetoid) and GCN (https://arxiv.org/pdf/1609.02907.pdf)
> > > > 2. The version used by the authors (https://github.com/EdisonLeeeee/GraphData). It has nodes 3,312, edges 4,715 and feature 3703.
> > > > 3. The earliest version of citeseer (described in https://linqs.soe.ucsc.edu/data).
> > > >
> > > > I made the following observations.
> > > > 1. Version 2 (https://github.com/EdisonLeeeee/GraphData) also states that it comes from Planetoid. Therefore, I believe that Version 1 and 2 are more or less the same dataset.
> > > > 2. I trace back to the earliest version of citeseer (version 3). I find out that in version 3 of citeseer, 3327 nodes are in the graph (citeseer.cites), but only 3312 nodes have attributes (citeseer.contents). This leads to the 3312 VS 3327 discrepancy. After removing the 15 nodes without node attributes/labels, the remaining 3312 corresponds to the citeseer Version 2 in terms of metadata and graphs. This further states that the citeseer versions 1, 2, 3 are the same.
> > > > 3. I implemented the workflow of the authors (20 nodes per class for training, 500 validation, else testing) using a 2-layer GCN. For both version 1 and 2, GCN gives ~65%-66% testing accuracy using bidirectional versions. This casts doubts on the reported 56% GCN baseline.
> > > >
> > > > Considering the observations, I suggest the authors to redo GCN experiments on bidirectional citeseer (and further, all baseline results copied from https://proceedings.neurips.cc/paper/2020/file/cffb6e2288a630c2a787a64ccc67097c-Paper.pdf).

---

> > > > > ### Author Response · Authors · 2022-06-20
> > > > > **Thanks**
> > > > >
> > > > > Thank you.   We will re-do most of baseline algorithm over the Version 2 citeseer dataset in the next week time as for some baseline we dont have their codes.

---

> > > > > ### Author Response · Authors · 2022-06-20
> > > > > **Quick test**
> > > > >
> > > > > We quickly did a test over citeseer (version 2) with torch geometric GCN.  Here is the result:
> > > > >
> > > > > Experiment 001 with seed  1000: Average test accuracy over 10 reps: 0.6167 with stdev 0.0089
> > > > > dataset: Citeseer; epochs: 200; reps: 10; learning_rate: 0.00500; weight_decay: 0.0050; nhid:  16; dropout: 0.70;
> > > > >
> > > > > We will update our Table with the new results.  It seems we cannot send through the python code here.

---

> > > ### Author Response · Authors · 2022-06-20
> > > **Clarification 3**
> > >
> > > We just did a quick test with torch geometric and found that citeseer from geometric is an undirected graph, so not sure whether the citeseer in https://arxiv.org/pdf/1609.02907.pdf is undirected.

---

> > > > ### Comment · Reviewer_9MKT · 2022-06-20
> > > > **Results after my checking**
> > > >
> > > > I did some checks myself and made the following observations.
> > > >
> > > > 1. GCN deals with bidirectional (undirected) graphs. It is clearly stated in the experimental setting of https://arxiv.org/pdf/1609.02907.pdf (Page 6).
> > > >
> > > > 2. In https://arxiv.org/pdf/1609.02907.pdf, citeseer is said to have 3327 nodes. However, 15 of them do not have node attributes (see the earliest citeseer https://linqs.soe.ucsc.edu/data). Removing these nodes leads to 3312 nodes, which is the same as https://github.com/EdisonLeeeee/GraphData.
> > > >
> > > > Further details can be seen in my reply to clarification 2.

---

> > > ### Author Response · Authors · 2022-06-20
> > > **Large-scale Experiment**
> > >
> > > Given the time constraint, we have not done Reddit yet.   Here is one point we would like for you to consider that the recipe is provided in Section 3.5 in handling larger graphs. In some cases, that $A^TA$ would be sparse enough for sparse matrix operations in the suggested algorithm, unless a larger graph has denser edges.  We are continuing to do this.

---

### Review · Reviewer_KiFZ · 2022-06-07

**Summary Of Contributions:**

Graph Neural Networks have been largely developed for undirected graphs. Recent works, however, have started looking to develop GNNs for directed graphs. This paper proposes a framelet based model similar to the one proposed in Zheng et. al [A] and Yang et. al [B] For directed graphs, it is non-trivial to extend classical laplacian based graph fourier transform approaches. Instead, the authors propose to define convolutions based on the singular values of the adjacency matrix (with self loops). The authors extend the quasi-framelet technique from [B] for the singular values and adapt the proofs from [B] to show it can reconstruct signals perfectly. The authors also propose modifications of the model which can scale to large graphs. The results demonstrate that the proposed approach does better than the state of the art models on directed graph dataset.

[A] How Framelets Enhance Graph Neural Networks, Xuebin Zheng, Bingxin Zhou, Junbin Gao, Yu Guang Wang, Pietro Liò, Ming Li and Guido Montúfar, ICML 2021
[B] Quasi-Framelets: Another Improvement to GraphNeural Networks, Mengxi Yang, Xuebin Zheng, Jie Yin, Junbin Gao, ArXiv 2022

**Broader Impact Concerns:**

Several social networks are actually directed graphs. For example, in Twitter, an edge from A to B indicates A follows B and it is not necessarily needed that B follows A. Development of models that allow to do improved predictions on such graphs can allow people to discover attributes of users in the social network which otherwise they were keeping private. Therefore, it would be good for the authors to add few lines regarding such impact.

**Requested Changes:**

1] There is a nice motivation in Introduction section about why SVD for directed graphs is a desired approach. I feel like that part can be trimmed down in the introduction and can be added in full detail in intro to Section 3, motivating the whole SVD-based approach.
2] Section 3.3 and 3.4 are hard to follow. I had to do multiple reads and I am still not fully convinced I understood what is being said in its entirety, I have a pretty good hunch what is being said though. It might be worthwhile to carefully rewrite these sections (particularly 3.4, since it is one of the contributions of the paper).
3] Comparison against UFG/QUFG: While it is clear that the method is targeting directed graphs, but it is still possible to compare against UFG and QUFG by simply converting the directed graphs into undirected graphs. Such a comparison will directly speak to the necessity of a such specialized models for directed graphs.

---
Update:
I went through the reviewer's notes on experimental number gaps between other papers reporting GCN numbers on citeseer versus the numbers reported in this work. The updated numbers still look far away from the earlier reported numbers. It is unclear as to what is the gap here.

**Strengths And Weaknesses:**

Strengths:
1] SVD-GCN. Extends the Quasi-Framelet model for directed graph based on SVD rather than eigenvalues of Laplacian
2] Improved Results. The proposed approach performs on par or better than SoTA models on various datasets

Weaknesses:
1] Clarity of writing. The paper could use some reorganization and more details to better understand the proposed ideas.
2] Experimental evaluation. The paper chooses to not compare against UFG and QUFG since it is working with directed graphs. However, same could be said for GCN etc. I am assuming that the graphs are made undirected to make it work with GCN. It is unclear as to why the same could not be done and UFG and QUFG could be run. It would answer the question: Can we simply make a directed graph into undirected, run UFG/QUFG and get improved performance or do we really need a specialized architecture like SVD-GCN that can directly operate on directed graphs to get improved performance?

---

> ### Author Response · Authors · 2022-06-20
> **Our response to your comments**
>
> 1. Regarding Section 3.3 and 3.4:
>
> First thank you for your concerns raised.  The confusion may come from the fact that we did not make the original equations (9) and (10) clear due to missing several components. Now we have revised them in the new version (highlighted). To better understand it, the formula can be regarded as the elementwise form for the matrices $\mathcal{W}$ and $\mathcal{V}$ in equations (5) - (8).  The purpose is to show we have similar quasi-framelet signals done shown in (Dong 2017) for the classic framelet.
>
> In Section 3.4, we can consider another reconstruction method. Instead of reconstructing into the left singular vector space, we reconstruct the signals simply in the right singular vector space $\mathbf V$.  We find in several experiments this way performs better.
>
> 2. Experiment Comparison with Framelet/GCN for undirected graphs
>
>     Regarding your question why SVD-GCN is not compared with UFG/QUFG but GCN, our answer is, basically GCN is a spatial message passing methood, it works for both undirected graphs and directed graphs straightaway. However UFG/QUFG is spectral methods for undirected graphs. As suggested by you, we have conducted UFG/QUFG algorithms over the five datasets. The results are shown in Table 2 in the updated version.   More details please refer to our answer to Reviewer mXnG.
>
> 3.  Reorganization and clarity of writing and presentation.
>
>     We have trimmed down a bit about the SVD-related motivation part in the Introduction section and move this part to Section 3 "SVD-Framelet Decomposition" as the first subsection named "The Motivation to apply SVD-based Approach". And we have also added more general motivation part in the Introduction section to support our main ideas.
>
>     We have addressed several typos and grammar errors and we are still revising the paper to increase its readability. Thank you for your suggestions and concerns.  A revised version has been uploaded for your convenience.

---

### Review · Reviewer_mXnG · 2022-06-08

**Summary Of Contributions:**

In this paper, the authors present SVD-GCN, a graph neural network for directed graphs. The model decomposes the normalized adjacency matrix of the directed graph using the SVD algorithm and then applies framelets over the dual orthogonal system defined by the SVD. To update the representations of the nodes, the model uses a filtering approach that is based on SVD framelets. The authors also present a more efficient variant which capitalizes on Chebyshev polynomials. The proposed model is evaluated on node classification datasets where in most cases it outperforms the baselines.

**Broader Impact Concerns:**

No Broader Impact Statement is required since the paper mainly provides methodological contributions.

**Requested Changes:**

As already discussed, one of my main concerns with this paper is that the proposed model is not properly motivated. It is not explained in the paper why SVD-GCN can generate high-quality representations for the nodes of directed graphs. For instance, it is not clear what properties of directed graphs can be captured by the proposed model that cannot be captured by previous models, and in what kind of setting we would expect SVD-GCN to outperform other baseline models.

In subsection 4.3, a large-scale experiment is performed where SVD-GCN is evaluated on cora_full that consists of approximately 19k nodes, 65k edges and fetaures of dimension 8k. In fact, I would not consider this as a large-scale experiment since the cora_full graph is not a very large graph (there exist much larger graphs in different repositories such as the Open Graph Benchmark). More importantly, it is mentioned in the paper that for graphs whose number of nodes is larger than 15k, training needs to be perfomed on CPU. This is a serious limitation of the proposed model especially since most real-world graphs contain hundreds of thousands or even millions of nodes.

The proposed model can produce graph-level representations if some pooling function is applied to the learned node representations. Thus, it would be nice if the authors also evaluated the proposed model on graph classification/regression datasets that contain directed graphs. I think that the ogbg-code2 dataset consists of such kind of graphs. However, the dataset is large and I am thus not sure whether the proposed model could scale to this dataset.

The writing and the flow of the paper leaves the reader with a feeling that this was an initial draft of the work. I would suggest the authors work on improving that. There are several typos and grammatical errors which the authors need to address, for instance:\
p.1: "sucess" -> "success"\
p.4: "they designs" -> "they design"\
p.4: "as a complex Hermitian matrices" -> "as a complex Hermitian matrix"\
p.4: "domain were used" -> "domain was used"\
p.4: "regulate other frequency" -> "regulate other frequencies"\
p.6: "noramlized adjacency" -> "normalized adjacency"\
p.9: "coral_full" -> "cora_full"\
p.9: "fullChen" -> "full (Chen"\
p.9: "nodes represents" -> "nodes represent"\
p.10: "achieves the similar performance" -> "achieves similar performance"\
p.10: "results demonstrates" -> "results demonstrate"\
p.10: "In this experiment, we wish...." -> sentence does not make sense\
p.11: "is normalized" -> are normalized\
p.11: "method still maintain" -> "method still maintains"\
p.12: "state-of-the-arts" -> "state-of-the-art"

It is not clear to me from the paper whether Chebyshev polynomials are used in the experimental evaluation or whether the graphs' normalized adjacency matrices are decomposed using the SVD algorithm. I would recommend the authors perform some experiment to compare the exact algorithm against the one that employs the Chebyshev polynomial approximation.

In subsection 4.4, the authors claim that they perform a denoising experiment. In my understanding, they inject noise to the node features and then train and evaluate the model on the noisy setting. Thus, I think that technically the term "denoising" is not correct. The authors study the robustness of the SVD-GCN model to noise. I would thus suggest the authors rephrase the related sentences.

In equation 14, the superscript of $g_\theta$ is wrong, i.e., $i$ should be replaced with $k$.

When citing some prior work, both authors and years should be surrounded by parenthesis, e.g., in p.1 "Wu et al. (2020)" -> "(Wu et al. 2020)"

**Strengths And Weaknesses:**

With regards to the proposed architecture, in my view, the paper seems to be proposing an incremental contribution for the graph representation learning community. Most existing message passing graph neural networks can directly handle directed graphs, while there have also been proposed several models for that kind of graphs. More importantly, it is not properly explained in the paper what makes the SVD-GCN model appropriate for processing directed graphs and what are its advantages over previous models. Another weakness of the paper is the low quality of writing/presentation. I would suggest the authors put more effort on polishing the paper, in view of numerous typos and grammatical errors (some are listed in the next section). Furthermore, I would expect a more extensive experimental evaluation of the the SVD-GCN model. The main strength of the paper lies in the model's empirical performance. As can be seen in Table 2, the model outperform the baselines in most cases, while in Table 4, it is shown that it is robust to noise.

---

> ### Author Response · Authors · 2022-06-19
> **Our Answer to your review**
>
> 1. Regarding the contribution of the paper:
>
> We strongly disagree with you on "The paper seems to be proposing an incremental contribution for the graph representation learning community".  Fundamentally, SVD is a very powerful tool in signal processing, e.g. see [Masaki Onuki, Yuichi Tanaka, and Masahiro Okuda. Improved eigenvalue shrinkage using weighted chebyshev polynomial approximation. In ICASSP, pp. 4541–4545. 2017]. As we mentioned in the end of Introduction,  this is the first attempt to introduce adjacency SVD for graph convolutions neural network. The SVD for signal analysis can be explained as (1) decompose the signals over the space defined by right singular vectors of the adjacency matrix, then (2) appropriately filtering can be conducted in this space, and finally (3) reconstruct the signal according to the left singular vectors (or its space). In the case of spectral GNNs for undirected, these two spaces are the same.
>
> The second contribution is that a UFG (framelet) alike filtering framelet analysis is established, see equations (5)-(8) in Section 3.2. This is an entirely new approach combining the SVD and Framelets (multiple scale filtering). The third contribution is that we build the theory that the new dual orthogonormal systems offered by SVD provide the guarantees of graph signal decomposition and reconstruction. The benefits demonstrated in all the experiments come from this theorectical guarantee.
>
> Finally all the experimental results demonstrate this is a new effective and powerful GCN model.
>
> 2. Regarding the necessity of SVD-GCN.
>
> Our idea is based on the following observation.  There exist a number of GNN models for directed graph, like the class GCN. They are doing good work. We also note that UFG/QUFG have demonstrated its significant improvement over the classic GCN etc. This enhancement comes from UFG/QUFG's capability of multiscale filtering. However UFG/QUFG relies on the fundamental conditions of SPD Laplacian, a property that has lost for directed graphs. How to revive or maintain this capability for directed graph prompts our idea to extend the spectrl decomposition of Laplacian into a dual orthogonal decomposition defined by SVD for any matrices. According to our research in this paper, it turns out the newly proposed model SVD-GCN can do much better over almost all the existing GNN models for directed graphs.
>
> 3. Regarding the clarity and presentation:
>
> We appreciate your detailed comments on grammatical issues. We have addressed all the typos and grammatical errors listed in the comments and we are proofreading the paper and planning to proofread after all contents are amended accordingly. In the rest of review period, we will do our best according to your suggestions to revise the manuscript to improve its readability and clarity.
>
> 4. The Changes:    We completed the following tasks in this short revision period.
>
> 4.1  Revised the Introduction and relevant sections to improve the clarity, and conduct a thorough proofread and fix any possible issues.
>
> 4.2  Conducted several experiments against UFG/QUFG (also as suggested by other reviewers) to further demonstrate the effective of SVD-GCN for directed graphs. The results are reported in Table 2 of the new version. Please note the major difference of UFG and QUFG is in applying different framelet functions. We take the best performers Linear (UFG) and Entropy (QUFG) in the experiments and report the best among the two.  The general observation is that using multiscale filtering does improve the node classification performance. We also note that particularly SVD-GCN performs much better in two Amazon datasets against the UFG/QUFG. Our interpretation is that the edge diction does matter.
>
> 4.3  We tried to conduct a large scale experiment within our resources. We did a try for Reddit dataset, but we met the memory issues for both SVD-GCN and UFG/QUFG.  Unfortunately we do not have sufficient computing resources as that in a big company. Having said that, we do wish to have your attention to the recipe provided in Section 3.5 in handling larger graphs. In some cases, that $A^TA$ would be sparse enough for sparse matrix operations in the suggested algorithm, unless a larger graph has denser edges.

---

### Decision · Action_Editors · 2022-07-22

**Recommendation:** Reject

**Comment:**

The paper attempts to improve graph neural networks (GNNs) for directed graphs. In this regards, propose to perform convolution over the spectral domain given by SVD of the directed adjacency matrix. Also the quasi-framelet technique from Yang et al. is extended. The empirical results in the paper seem to demonstrate that the proposed approach does better than the state of the art models on directed graph dataset. However, the reviewers have concerns about the empirical results: 1) important baselines are missing compared to which reported numbers might not be strong (e.g. on citeseer), 2) the importance of capturing directionality of the graph is not clear as undirected approaches seem to match/exceed performance, and finally 3) missing large-scale experiments. Some reviewers found the motivation of the paper unclear. Furthermore all the reviewers are in consensus that writing and presentation of the paper need improvement. Thus, unfortunately I cannot recommend an acceptance without another round of review of the paper.

Improving the presentation and running a more thorough evaluation suite will make the paper useful for the community.